# Association with Different Housing and Welfare Parameters on Results of a Novel Object Test in Laying Hen Flocks on Farm

**DOI:** 10.3390/ani13132207

**Published:** 2023-07-05

**Authors:** Jennifer Hüttner, Annette Clauß, Lea Klambeck, Robby Andersson, Nicole Kemper, Birgit Spindler

**Affiliations:** 1Institute for Animal Hygiene, Animal Welfare and Farm Animal Behaviour, University of Veterinary Medicine Hannover, 30173 Hannover, Germany; annette.clauss@wki.fraunhofer.de (A.C.); nicole.kemper@tiho-hannover.de (N.K.); birgit.spindler@tiho-hannover.de (B.S.); 2Department of Animal Husbandry and Poultry Sciences, University of Applied Sciences Osnabrueck, 49090 Osnabrueck, Germany; l.klambeck@hs-osnabrueck.de (L.K.); r.andersson@hs-osnabrueck.de (R.A.)

**Keywords:** animal welfare, novel object, laying hens, feather pecking, cannibalism

## Abstract

**Simple Summary:**

In this on-farm study, 16 flocks in Germany were monitored for plumage and skin conditions as indicators of feather pecking and cannibalism. Furthermore, their behavior was monitored in a novel object test. During the laying period, the monitored flocks were visited four times. The novel object test can be used to assess the fearfulness of a flock. The average number of hens gathered around the novel object gives an indication of the fearfulness of a flock. The study indicates that an increase in fearfulness of the flock is associated with an increase in feather damage and cannibalism. Fearfulness also increased with hen age and flock size. White hens showed greater fearful behavior in contrast to brown hens. Hens on barn farms showed a higher fear response than hens in free-range farms.

**Abstract:**

The objective of this on-farm study was to determine if flocks showing feather damage and/or cannibalism would have a higher fear response to the novel object (NOT) and the association between different housing and welfare parameters on results of the NOT. Therefore, 16 flocks were observed during the laying period in Germany. In total, there were six barns, seven free-range, and three organic flocks. The plumage and integument condition of 50 birds of each flock were evaluated at 4 different times during the laying period (V1: 18th to 23rd week of life, V2: 26th to 35th week of life, V3: 49th to 57th week of life, V4: 61st to 73rd week of life). At the same observation times, the NOT was performed in the flocks (at four different locations per visit time). Based on the average number of hens gathered around the novel object (NO) within the period of two minutes, conclusions can be made about the fearfulness of the flock. The present study shows that the more fearful a flock was, the more frequent feather damage (*p* < 0.001) and cannibalism (*p* < 0.01) occurred. Age and flock size were associated with fearfulness. Fearfulness of hens increased with the increasing age of hens (*p* < 0.001) and with an increasing flock size (*p* < 0.001). Hens of white feather color appeared to be more fearful than brown hens (*p* < 0.001). Hens kept on barn farms showed significantly lower numbers of hens around the novel object (*p* < 0.001) than on free-range farms, which possibly indicates that having permanent access to outdoor space generally appears to be associated with the fearfulness of a flock.

## 1. Introduction

Animal welfare is an essential, challenging factor in the current intensive use of all farm animals, which must be improved [1,2,3,4]. Fear is an emotional state which is triggered by the awareness of danger or a potentially menacing situation that may be detrimental to welfare [5]. This fear behavior is strongly, negatively correlated with exploratory behavior [6,7]. Confronting animals with a novel stimulus leads to approach-avoidance conflict [8]. Thus, both exploratory behavior necessary for survival (e.g., foraging behavior) and avoidance behavior in potentially threatening situations are stimulated [8]. Different methods to measure this emotional state in animals are described in the Welfare Quality Assessment Protocol [9], which was written for pigs, cows, and poultry and, since 2019, for laying hens. It serves as an animal welfare assessment system to measure certain animal-related data, such as body condition of animals, health aspects, injuries, or behavior, which are taken into account [9].

Excessive fear can lead to reduced welfare and thus, to lower performance in poultry flocks [10,11]. Previous research has tried to establish a connection between fearfulness and the tendency for feather pecking (FP) and cannibalism in a flock under experimental conditions [12,13]. A higher fear response could be associated with an increased occurrence of feather pecking in laying hens [14,15,16]. Johnsen et al. [13] observed that in groups of red junglefowl housed on wire, they were more fearful than those housed on sand and straw at 42 weeks of life, and the birds also showed increased FP at the 45th week of life. Vestergaard et al. [12] identified an association between fearfulness and dustbathing behavior in groups of red junglefowl. Hens that showed the most FP were also the most fearful and demonstrated the least dustbathing behavior in their group. Both studies were conducted under experimental conditions and groups of hens were created for this purpose.

It has also been found that the relation between FP and fear is related to the age of the hens [17]. Hocking et al. [17] showed that laying hens avoided new objects less with increasing age. The older the birds, the more they exhibited behaviors such as foraging and pecking at feather bunches. In a study by Albentosa et al. [18], hens also showed a decrease in fearfulness with increased age. In both studies, open field tests, novel object tests, and tonic immobility tests were used as behavioral tests in hens kept under experimental conditions.

Flock size may also be associated with the occurrence of feather pecking [19,20]. For example, experimental studies by Nicol et al. [19] and Bilčik and Keeling [20] showed higher levels of feather pecking with an increasing flock size. In addition, as flock size increases, the level of fear also increases [21].

The novel object test (NOT) is one of several behavioral tests to assess the fear and exploration of novel objects in animals [9]. This test is applied in poultry as in other animals, for example, cattle, pigs, sheep, and horses [22]. For example, in the study by Dalmau et al. [23], 18 commercial farms with growing pigs were investigated. The novel object test was performed on these farmed pigs using three different colored balloons. It was found that recording contact latency, the time it takes for an animal to come into contact with the novel object (a balloon), is the best method for recording fearfulness [23].

The objective of this on-farm study was to determine if flocks showing feather damage and/or cannibalism would have a higher fear response in the novel object test. In addition, the association with feather color, age, flock size, and the husbandry system on the novel object test were identified. For this purpose, 16 flocks with different feather colors were examined in indoor, free-range, and organic housing systems.

## 2. Materials and Methods

### 2.1. Animals and Husbandry

For this on-farm study, 16 laying hen flocks with intact beaks in cage-free housing located in north-west (Lower Saxony) Germany were involved. All flocks were housed and managed under practical conditions in accordance with EU [24] and national law [25] and national requirements for animal husbandry [26]. The flock controlling was guided by the management recommendations of the respective breeding companies [27]. All housing systems were equipped with an aviary system during rearing and the laying phase (different types). The flocks were provided with standard commercial or organic feed (structured mash feed with a phase-feeding-program: Pre-lay, Layer Phase 1, Layer Phase 2, Layer Phase 3) and a common feeding regime (fed ad libitum with up to five feeding times), as well as a usual lighting program (14 h light and 10 h dark) and standard litter material (wood shavings, straw pellets, or lignocellulose pellets) and enrichment material (alfalfa hay and/or pecking blocks). The hens were vaccinated in accordance with standard industry procedures, e.g., vaccinations against Newcastle disease and Marek disease.

The flocks were visited at four different times during the laying period (V1–V4). These visiting times were set at the beginning, in the middle, and at the end of production to provide the best possible overview of the complete laying period. The first visit took place between the 18th and 23rd week of life (V1). V2 was between the 26th and 35th week of life, V3 was between the 49th and 57th week of life, and V4 was between the 61st and 73rd week of life. At all visits, the novel object test (NOT) as well as feather and integument scoring from a representative number of hens were carried out. The results represent an average for the four different visiting times.

In compliance with the European Directive [28], the experiment did not include any invasive treatment of the hens.

Table 1 shows the number of involved flocks under consideration of selected parameters. Flock size was categorized as extra small (up to 1500 animals), small (more than 1500 up to 5000 animals), medium (more than 5000 up to 10,000 animals), and large (more than 10,000 animals). The different genetics were summarized into the feather color brown (B) and white layers (W). Hens housed in barns without access to free-range (hereafter referred to as “barn farms”) were kept in indoor housing, including the opportunity to access a winter garden, which is an area that was protected from wind and weather and had natural light, for some of them. Hens housed in free-range farms had permanent access to the outside. Additionally, organic farms offered a lower stocking density of a maximum of six hens/m^2^ instead of a maximum of nine hens/m^2^ in barns or free-range farms. Hens were kept on the farms until they were 62 to 95 weeks old, having been raised in aviaries until 16/17 weeks of age.

### 2.2. Novel Object Test

The novel object test (NOT) was used as a common tool to investigate the fearfulness of hens. The NOT was conducted according to the welfare quality protocol [9] at all visits (V1 to V4). The novel object (NO) used in this test was a 50 cm long stick, which was covered with colored stripes (Figure 1).

The NOT was conducted by one and the same observer before the plumage and integument scoring was performed. The same object was used for all four visit times. After five minutes, when the animals had had the opportunity to acclimatize to the observer and to follow their normal behavior, the observer placed the NO in the litter. According to the Welfare Quality Poultry Protocol [9], the observer moved about 1.5 m away from the NO in order to avoid influences on hens’ behavior by the observer. Immediately, the number of hens gathering around the NO was counted every ten seconds for a total time of two minutes. Only a radius of “one hen’s length” around the NO was taken into account.

The NOT was performed at four locations in the litter, representative of the distribution of the flock in the barn (front, middle, and back of the barn) [9]. Thus, the NOT was conducted a total of 256 times in the 16 flocks (on average, 16 times per flock) and during the light period after the main laying time of the hens. Furthermore, there was no feeding performed during the performance of the NOT.

### 2.3. Feather and Integument Scoring Method

A scoring method containing the handling of individual animals (Hands-on Scoring, HSc) was performed to assess the plumage and integument condition of the hens in every flock. The HSc contained a sample of at least 25 to 50 hens per visit. This number mainly depended on the level of unrest in the barns. If a herd was very restless, 25 animals were caught in order to cause the animals as little stress as possible. In each compartment of a barn, hens were assessed in the littered scratching area, in the center of the aviary, and in the upper area of the aviary to obtain a representative sample of the flock. It was not possible to perform the HSc with a sufficiently high number of animals (25 to 50) in all visits (two flocks in V1, one flock in V2, V3, and V4). The reasons for this were, for example, high temperatures during the summer period which led to a nervous flock. The latter made it impossible to catch individual animals from the flock without putting the hens under high stress. In this case, the results from a visual assessment (Visual Scoring, VSc) were used. This is a valid alternative to the HSc to assess the integument and plumage damage [29]. For the VSc, a sample of at least 120 to 160 hens was scored in the same locations as the HSc. Two trained observers had previously performed an inter-observer reliability, which was calculated on the basis of 50 (HSc)–120 (VSc) hens (Table 2).

In both assessments, five body areas were considered: head/neck, rump, wings, tail, and belly [30]. A four-point scale was used to score the body regions for plumage and integument condition (Table 3). The severity of the feather loss was determined by a gradual classification.

Based on the number of animals affected with feather losses or skin lesions, each flock was categorized and classified according to the categories 0 to 3, as shown in Table 4.

### 2.4. Statistical Analyses

Statistical analyses were performed using the SPSS program (version 29.0). The inter-observer reliability was performed for the feather, and integument scoring and was calculated using Krippendorff’s alpha. The categorization was completed according to the suggestion of Landis and Koch [31]: (<0.00 = poor, 0.00–0.20 = slight, 0.21–0.40 = fair, 0.41–0.60 = moderate, 0.61–0.8 = substantial, 0.81–1.00 = almost perfect). Means and standard deviations (±) of the number of hens at the NOT were calculated for each flock and visit.

The NOT data were analyzed by the Kruskal Wallis test and paired comparisons. The average number of hens gathered around the novel object (fixed factor) was compared to the housing system, age, flock size, feather damage, cannibalism, and feather color (all random factors). For statistical evaluation, different genetics were summarized into the feather color brown (B) as well as white layers (W). The level of significance was set to *p* < 0.05.

## 3. Results

### 3.1. Housing System

Figure 2 shows the association with the different housing types of barn, free-range, and organic on the average number of hens gathered around the novel object for all visits (16 evaluated NOT per flock). With an average of 3.1 (standard deviation (SD) 3.3) hens on the barn farms (n = seven farms), significantly (*p* < 0.001) fewer hens were counted at the NO during the NOT compared to the free-range farms (n = seven farms). On organic farms (n = three farms), a mean of 4.03 (SD 5.1) hens and, on the free-range farms, 3.6 (SD 2.9) hens were gathered around the object (not significant (n.s.); *p* = 0.331 and *p* = 0.058).

### 3.2. Age

Figure 3 shows the results of the NOT at the different visits (age of the hens at the different visit times V1–V4). In general, the number of hens in the NOT decreased significantly with increasing age (*p* < 0.001) up to visit V3.

At the first visit (V1; 18th to 23rd weeks of life), an average of 7.06 (SD 4.7) hens were detected around the novel object, whereas at V2 (26th to 35th weeks of life), less than half of the hens compared to the first visit (*p* < 0.001) were documented at the NO (average of 2.7 (SD 2.1) hens). At visit 3 (49th to 57th weeks of life) and visit 4 (61st to 73rd weeks of life), nearly the same number of hens were gathered around the novel object, with an average of 1.97 (SD 1.7) and 2.20 (SD 2.0) hens, respectively (n.s.; *p* = 0.13).

### 3.3. Flock Size

The association between flock size and the NOT in the 16 involved flocks is shown in Figure 4. In the extra small flocks, the average number of hens gathered around the novel object was significantly higher, with an average of 5.52 (SD 4.6) hens compared to that observed in the small (*p <* 0.001), medium (*p <* 0.001), and large flocks (*p <* 0.001). With 3.79 (SD 4.1) hens gathered around the novel object, the small flocks showed a significantly higher number of hens gathered around the object compared to medium flocks (2.71 hens, SD 2.4, *p <* 0.001) and large flocks (2.19 hens, SD 2.4, *p* < 0.001). In addition, there were significantly more hens gathered around the object in the medium flocks (2.71 hens, SD 2.4) compared with the large flocks (2.19 hens, SD 2.4, *p* < 0.001).

### 3.4. Feather Damage and Cannibalism

The number of hens gathered around the novel object in the context of the categories of feather damage in the flock is shown in Figure 5. In general, the presence of feather damage in the flock is significantly associated with the results of the NOT (*p* < 0.05). The following figure indicates that with 6.2 (SD 4.7) hens, there were more animals gathered around the object on average where no feather damage (category 0) was documented than in flocks where slight feather damage (category 1, 3.5 hens, SD 3.4, *p* < 0.001), moderate feather damage (category 2, 2.4 hens, SD 1.8, *p* < 0.001), or severe feather damage (category 3, 2.1 hens, SD 1.9, *p* < 0.001) was present. On average, there were 3.5 hens (SD 3.4) gathered around the object in flocks with slight feather damage (category 1). This was significantly more hens compared to flocks with moderate feather damage (category 2, 2.4 hens, SD 1.8, *p* < 0.01) or severe feather damage (category 3, 2.1 hens, SD 1.9, *p* < 0.001). If moderate feather damage (category 2) occurred in the flocks, there were significantly more hens (2.4, SD 1.8) gathered around the novel object than in flocks with severe feather damage (category 3, 2.1 hens, SD 1.9, *p* < 0.01).

Figure 6 shows the association between cannibalism in the flocks and the NOT. If there was no cannibalism (category 0) in the flock, there were, on average, 3.7 (SD 3.7) hens gathered around the novel object. There were, on average, significantly more hens around the object than in flocks with slight cannibalism (category 1, 3.1, SD 2.6, *p* < 0.05) or massive cannibalism (category 2, 0.7 hens, SD 0.7, *p* < 0.001). In the presence of slight cannibalism (category 1) in the flock, there were, with 3.1 (SD 2.6) hens, more animals gathered around the object in comparison to flocks where massive cannibalism (category 2, 0.7 hens, SD 0.7, *p* < 0.001) occurred.

### 3.5. Feather Color

The number of hens gathered around the novel object in the context of feather color is shown in Figure 7. The results indicate that during the entire production period, an average of 4.0 (SD 3.8) hens of brown feathered flocks and an average of 2.2 (SD 2.7) hens of white feathered flocks were documented as being gathered around the novel object (*p* < 0.001).

## 4. Discussion

The objective of this study was to indicate the association with different housing and welfare parameters on the results of a novel object test under field conditions. Fearfulness, as indicated in this study by the response to a novel object, was investigated under the association with feather damage and cannibalism as well as feather color, age, different flock size, and housing systems. This study indicates that an increase in fearfulness is associated with an increase in feather damage and cannibalism in a flock. Fearfulness also increased with the age of the hens and flock size. White hens behaved more fearfully than brown hens. Hens on barn farms showed a higher fear response than hens in free-range farms. In comparison to previous studies, the present investigations involved 16 monitored flocks, a larger number of commercial flocks conducted under practical conditions. It should be critically considered that the flocks were only visited at four visiting times. More intensive monitoring and thus, more frequent visits would show a more detailed progression of behavior. In addition, the flocks within this study differed greatly from one another due to their feather color and management.

In general, the results explore that during the first visit (V1, 18th–23rd week of life), all flocks showed, on average, a lower fear response compared to the other visits during production (V2: 26th–35th week of life, V3: 49th–57th week of life, V4: 61st–73rd week of life), as indicated by a higher number of hens being gathered around the novel object. This suggests that the animals had developed more fearfulness with age and thus, may have been more stressed than at the beginning of the laying period, although rehousing of pullets and new management conditions at the laying farm are also stressful for the hens. In addition, it is questionable whether acclimatization of hens to the object is a factor which is causing a decrease in interest. Another factor to consider is that hens may show less explorative behavior with age. This result of the present study differed in certain points from the studies by Hocking et al. [17] and Albentosa et al. [18], where the hens showed a decrease in fearfulness with increasing age. However, it should be taken into account that the times for performing the test were at an earlier age in both studies (Hocking et al. [17]: 0–31 weeks of life; Albentosa et al. [18]: 4–12 weeks of life) compared to our study. Thus, the hens in these studies were considerably younger than in the present study and were monitored for a much shorter period of time, which may have affected the fearfulness of the flocks. In addition, the studies were conducted under experimental conditions and not on commercial farms, as in the current study.

The results show that as flock size increases, there were significantly fewer hens gathering around the novel object. This allows, in our study, the conclusion that hens in smaller flocks show a lower fear response than animals in larger flocks. The study of Bilcik et al. [21] also showed an increase in fearfulness with an increase in flock size, taking into account that the hens in that study were kept under experimental conditions and in group sizes of 15, 30, 60, and 120 birds. In general, it can be assumed that the behavior of small (e.g., experimental) groups differs from that of larger groups. Potentially, hens in smaller flocks come into more contact with humans during routine daily inspection than those in larger flocks. Thus, they become accustomed to human contact and react less fearfully. In general, if the flocks are inspected more frequently, they can get used to the intense contact and may therefore show less fearfulness. Studies show that there may be an association between the behavior of livestock keepers toward their animals and their impact on the behavior, welfare, and productivity of farm animals, as it may cause fear-related stress [32,33]. The intensity of care is also important for the conditions in rearing to accustom animals to people.

The novel object test shows that flocks which have a higher category of feather damage or cannibalism correlate with a higher fear response. The number of hens gathered around the object decreases significantly in flocks with slight feather damage or a severe grade of cannibalism. As already found in other studies, an increased fear response was correlated with an increasing tendency to feather pecking [14,15,16]. Birds with higher levels of fearfulness are more prone to severe feather pecking [15,34]. As already reported in other studies, these severe feather pecks can lead to cannibalism [35]. Thus, if slight forms of feather damage already occur, action should be taken to avoid them resulting in cannibalism. It is possible that this slight feather damage could be detected at an early stage using the NOT. This result shows that flocks where feather damage and cannibalism occur are more fearful of a novel object. In the study presented here, the flocks were visited only at four visiting times during production, up to the 73rd week of life. More intensive monitoring and thus, more frequent visits would show more detailed progression of behavior. However, this would have to be tested by additional studies at shorter visit intervals to capture a change in behavior even more accurately. It should also be considered that management interventions may also have an impact on the NOT when feather pecking and cannibalism occur. Management strategies, such as adding enrichment or additional loose litter material, may have a positive influence on the behavior of the birds, which could then be shown in the behavioral test by an increase in the number of hens gathered around the object. In addition, it should be taken into account that the NOT may be influenced not only by the fear behavior, but also by the exploration behavior of the hens [6,7]. Thus, there is a possibility that the animals are confronted with an approach-avoidance conflict due to the NO [8]. It is conceivable that some hens differ in their exploratory behavior and not only in fear.

With an average of 2.2 hens, there were significantly fewer white hens gathered around the novel object compared to brown hens (average 4.0). Thus, the results of an NO indicate that white hens are more fearful than brown hens. However, it should be noted that only five flocks with white colored feathers could be included in this study. Nevertheless, other studies show comparable results [10,16,36]. Jones [10] already showed that hens of white genetics show a longer tonic immobility [10]. This may be associated with higher fearfulness compared to brown genetics [18,36]. Uitdehaag et al. [16] found that hens of white strain White Leghorn (WL) were more fearful in behavioral tests than hens of brown strain Rhode Island Red (RIR). The more fearful WL showed more feather pecking in that study than the brown RIR [16]. From the first scoring time at 35 weeks of age, worse plumage was observed in the WL [16]. This could also have led to increased fear.

On barn farms, the mean number of hens gathered around the object was significantly lower than on free-range farms. The hens’ actual usage of the provided outdoor access was not recorded in this study. Therefore, to be able to draw further conclusions, this would have to be included in the data, as other studies already showed that this can have a positive effect on flock health [37,38]. In the aforementioned study, Green et al. [37] found that the risk of feather pecking decreased by a factor of 5 when at least half of the animals in a flock used the outdoor area on a bright, sunny day. It was hypothesized that by using the outdoor area, the stocking density in the barn and the perceived flock size would be lower, and the greater variety of pecking opportunities would result in a reduction in the incidence of feather pecking [37]. In a study from the United Kingdom, permanent access to an outdoor area is shown to be positive for hens because of the environmental enrichment and results in less feather pecking [39]. Studies regarding stocking density also found that lower stocking densities resulted in less feather pecking in flocks [19,40]. Additionally, structural modifications in aviary systems enhance the welfare of hens [41]. Laying hen producers may therefore adopt some of these structural modifications in aviary systems to enhance the welfare of hens [41].

Further studies are needed to capture the latency in the NOT. This would allow detailed conclusions to be drawn concerning fearfulness in the future.

## 5. Conclusions

The NOT was conducted a total of 256 times on 16 laying hen farms in Germany. The results show that in this study, feather damage and cannibalism were associated with fearfulness. White feathered hens showed higher levels of fearfulness compared to brown hens. Higher age and larger flock size also showed an increase in fear behavior. Having permanent access to outdoor space could be associated with fearfulness. Thus, more research is needed in this area, and there should be more intensive observations of the flocks and more visits. This would allow for better identification of the progression of hen behavior, resulting in earlier detection of feather pecking, if necessary. In addition, the latent period of the NOT, i.e., the time until the first animal enters the novel object, should also be recorded. This can be used to draw more detailed conclusions about the fear behavior of a flock, as this would show more behavioral differences in the various flocks. Furthermore, the exploration and avoidance behavior and their influence on the NOT should be considered.

## Figures and Tables

**Figure 1 animals-13-02207-f001:**
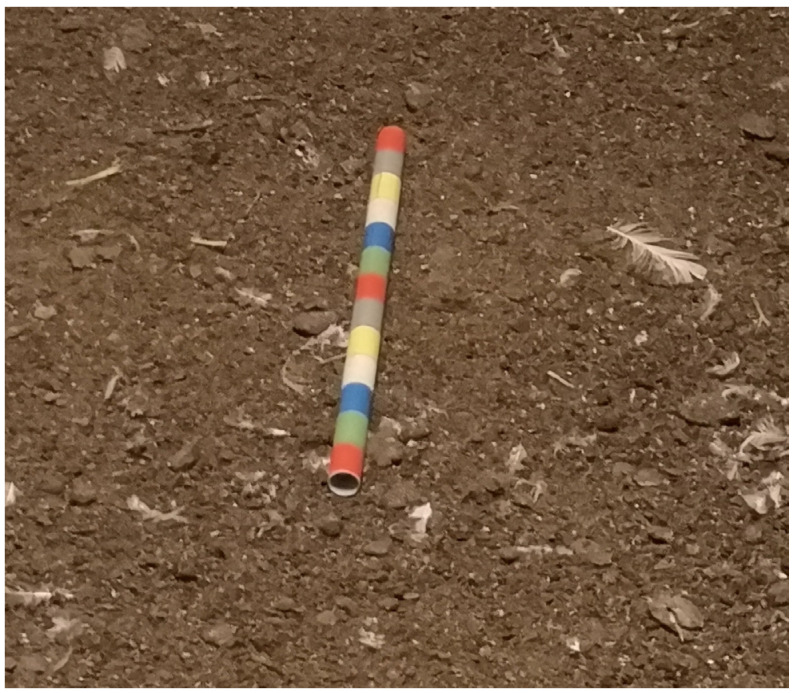
The novel object (NO). A 50 cm long stick covered with colored stripes was placed in the littered area of the barn. The number of hens gathering around the NO was counted every ten seconds for a total of two minutes.

**Figure 2 animals-13-02207-f002:**
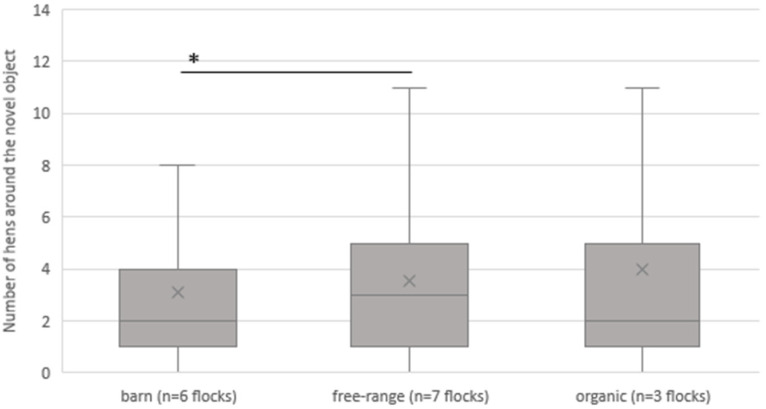
Average number of hens gathered around the novel object (mean of four visits of each flock) depending on the housing system. * = *p* < 0.001.

**Figure 3 animals-13-02207-f003:**
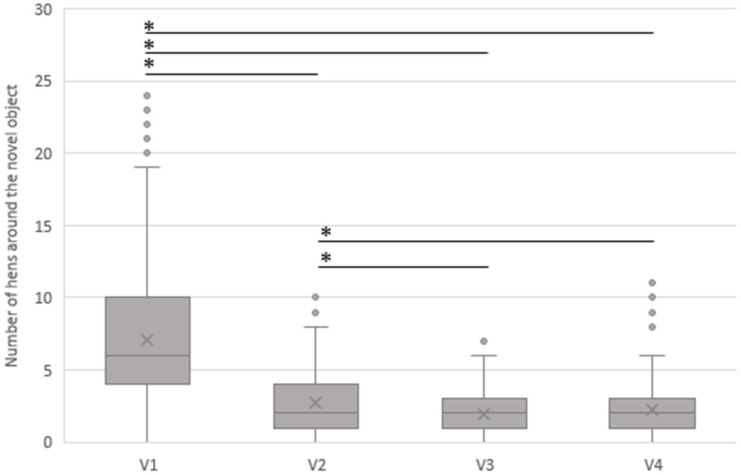
Association between the average number of hens gathered around the novel object and the age of the hens on the different visiting dates (V1–V4). V1: 18th–23rd week of life, V2: 26th–35th week of life, V3: 49th–57th week of life, V4: 61st–73rd week of life. Six flocks in barn systems, seven free-range flocks, and three organic flocks were evaluated. * = *p* < 0.001.

**Figure 4 animals-13-02207-f004:**
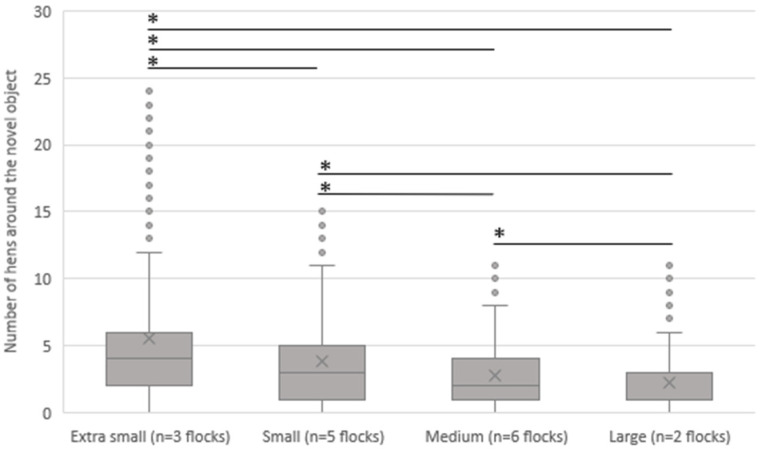
Association between the average number of hens gathered around the novel object (mean of four visits of each flock) and the flock size. Extra small: up to 1500 hens, small: more than 1500 up to 5000 hens, medium: more than 5000 up to 10,000, large: more than 10,000 hens. * = *p* < 0.001.

**Figure 5 animals-13-02207-f005:**
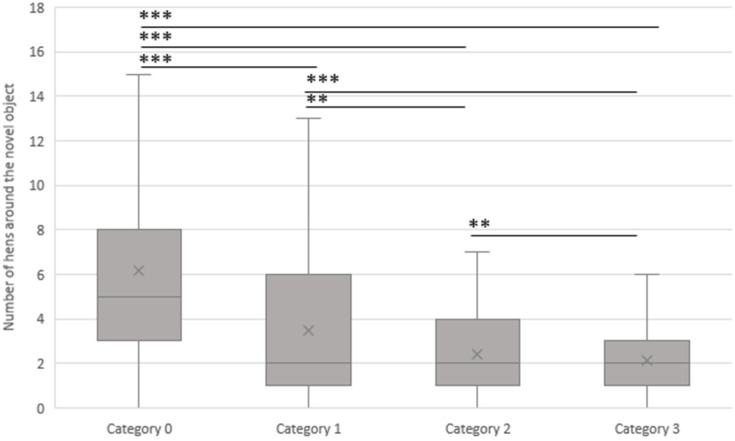
Association between the average number of hens gathered around the novel object and the feather damage categories (0 (no feather damage in the flock, n = 13), 1 (slight feather damage in the flock, n = 9), 2 (moderate feather damage in the flock, n = 9), and 3 (severe feather damage in the flock, n = 14)). The novel object test was conducted at the same visiting times as the scoring method in all 16 laying hen flocks. ** = *p* < 0.01, *** = *p* < 0.001.

**Figure 6 animals-13-02207-f006:**
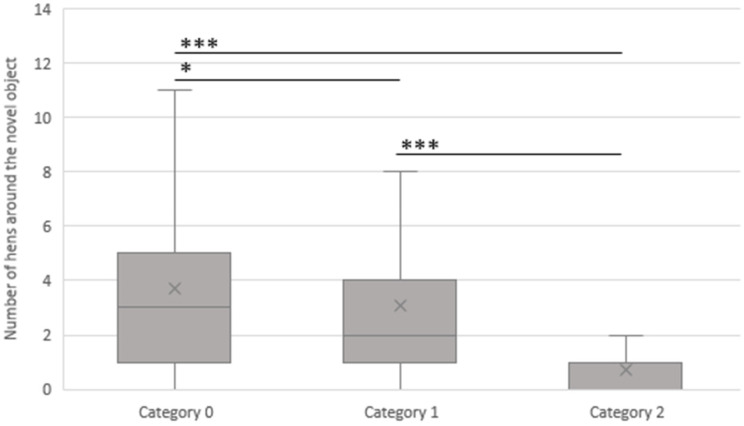
Association between the average number of hens gathered around the novel object and the cannibalism categories (0 (no cannibalism, n = 15), 1 (slight cannibalism, n = 6), and 2 (massive cannibalism, n = 2), (mean of four visits of each flock)). The novel object test was conducted at the same visiting times as the scoring method in all 16 laying hen flocks. * = *p* < 0.05, *** = *p*< 0.001.

**Figure 7 animals-13-02207-f007:**
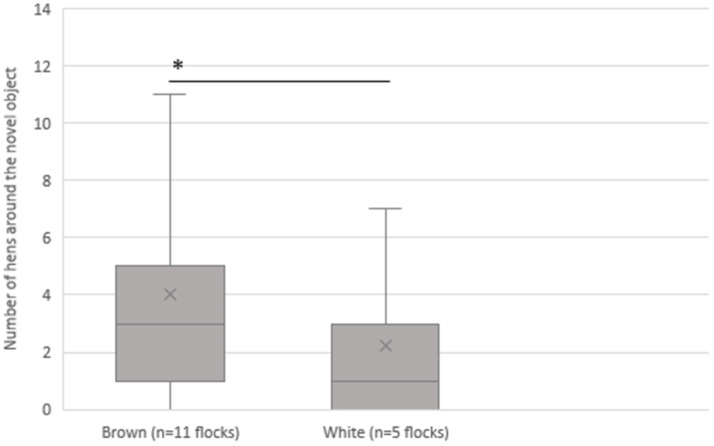
Association between the average number of hens gathered around the novel object and the different feather color of the hens. * = *p* < 0.001.

**Table 1 animals-13-02207-t001:** Description of the investigated 16 laying hen flocks with housing system, feather color, flock size, and flock number.

Parameters		Number of Flocks	
Housing System	Flock Size	White	Brown
Barn	Extra small (<1500)	1	-
	Small (1500–5000)	-	2
	Medium (5000–10,000)	2	1
	Large (>10,000)	1	-
Free-range	Extra small (<1500)	-	1
	Small (1500–5000)	-	3
	Medium (5000–10,000)	1	1
	Large (>10,000)	-	-
Organic	Extra small (<1500)	-	1
	Small (1500–5000)	-	-
	Medium (5000–10,000)	-	1
	Large (>10,000)	-	1

**Table 2 animals-13-02207-t002:** Description of the inter-observer reliability, which was calculated on the basis of 50 (HSc)–120 (VSc) hens for the integument and plumage damage. The Krippendorff’s alpha coefficient is indicated for the scored body regions: head/neck, rump, wings, tail, and belly.

Krippendorff’s Alpha Coefficient for the Scored Body Region	Integument Damage	Plumage Damage
Head/neck	1.0	0.61
Rump	1.0	0.74
Wings	1.0	0.41
Tail	0.73	0.70
Belly	0.40	0.73

**Table 3 animals-13-02207-t003:** Description of the scoring scheme used for the plumage and skin condition assessment of each scored hen, which forms the basis for the development of the categories of feather losses and cannibalism ([29], modified).

Score	Feather Losses	Score	Skin Lesions
0	No feather losses	0	No skin injuries
1	≤25% feather loss (slight damage)	1	Injuries of <1 cm diameter
2	>25% to ≤50% feather loss (moderate damage)	2	Injuries > 1 cm–<2 cm diameter
3	>50% feather loss (severe damage)	3	Injuries > 2 cm diameter

**Table 4 animals-13-02207-t004:** Description of the flock-category scheme used for categorizing the occurrence of feather damage and injuries in the flock based on the scored hens.

Flock-Category	Definition of Feather Damages	Flock-Category	Definition of Cannibalism
0	No animal received the score 1 or worse	0	No scored hen received the score 1 or worse
1	>15% of the scored hens received score 1 and worse (slight feather damages)	1	>1 scored hen received score 2 and worse (slight cannibalism)
2	>50% of the scored hens received score 1 and worse (moderate feather damages)	2	>15% of the scored hens received score 2 and worse (severe cannibalism)
3	>80% of the scored hens received score 1 and worse (severe feather damages)	3	-

## Data Availability

The data presented in this study are available on request from the corresponding author. The data are not publicly available due to privacy restrictions.

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
