# Peer review of "Association with Different Housing and Welfare Parameters on Results of a Novel Object Test in Laying Hen Flocks on Farm"

_animals, 2023, doi:10.3390/ani13132207_

Round 1

Reviewer 1 Report

Very interesting work! This is a well-conducted study that can be a valuable contribution to the literature after some changes to the language and maybe a closer look at the statistical analysis. I have some major comments and a few minor comments. It might look like a lot, but I believe that they will improve your manuscript to be even better.

Major comments:

Title, Abstract. Effect of welfare parameters on NOT indicates causality… maybe reword it to say that you investigated the association of welfare parameters and NOT.

Abstract

Give exact P-values not just <0.05

L33-35 Why would a difference in NOT not indicate a difference in fearfulness when it comes to housing effects? And if it does (as indicated by the data, though opposite to what you are saying here!) then you’ll need to change L 18-19 in the summary to reflect that.

Introduction

L39-40 Fear is triggered by the awareness of danger, not the other way around.

L53-57 Why is this section about pigs? There are good examples in the literature about novel object tests in laying hens. Even validations of the test in poultry, see https://doi.org/10.1016/j.physbeh.2007.03.016.

L57-58 In pigs or laying hens or in general? Please add the species you are discussing.

L51-77 This paragraph seems like a mix of too many things and should be split into multiple shorter paragraphs. One on fearfulness in laying hens, one on severe feather pecking (why is it a problem) and it’s relationship with fearfulness, and one the novel object test.

M&M

Please add the number of your animal use ethics approval and the agency that reviewed and granted it. Is it the Working Group ‘Laying Hens’ of the Animal Welfare Program of Lower Saxony that you mention on lines 88-89? If not, please explain what that is or delete this statement.

L97 Please indicate how many flocks had pecking stones and what group they belonged to. There is some literature suggesting that pecking blocks might reduce feather pecking which could be relevant to your study.

Table 1 it is unclear from this table whether there are any confounds between groups (e.g., if all organic farms housed brown birds). I suggest re-organizing it to show the factor combinations. For example, by having columns for browns and whites and rows with housing systems with flock size as a sub-category.

L155 a Krippendorff’s alpha coefficient of 0.4 is pretty bad. Which variable was that? Please provide the reliability scores for each variable individually.

L174 and following: Why did you use a non-parametric test? And why were your comparisons paired? Paired by what, you did not have a before and after treatments. Did you correct for multiple testing in the pair-wise comparisons? I would suggest consulting a statistician or if you did, then add some explanation. Some of you results seem off, e.g., how can the number of hens around the NO in V2-V4 be different if the boxplots completely overlap?

Figures 2-4 I would suggest merging these plots into a single figure as they describe single main effects each and therefore do not need to be too big. Also, you describe the mean and SD in the text and in a figure, which seems unnecessary. At least merge them into Figure 2A, 2B, 2C.

L195 and many more occasions afterwards: p<0.000 must be a typo. Please search for it and correct it everywhere.

L203 interaction is the wrong term, you did not find an interactive effect (did you even test for that?).

Figure 4 could the effect be driven by outliers?

L221 your sub header suggests causality. Did the feather damage make them more fearful? You cannot tell either way based on you study design. Just say association or reduce your sub headers to the factor, e.g., ‘feather damage and cannibalism’

Discussion

Add a summary of your findings at the beginning of your discussion.

L265-268 re-phrase: “Fearfulness as indicated by the approach of a novel object and feather pecking were investigated under the influence of different flock size, ...” The way you are stating it suggests that you measured fear (which you cannot, only its indicators) and that feather pecking is an influencer of fearfulness, which could just as well have been the other way around.

L278-280 This decrease of hens approaching the NO can have many different explanations in addition to an increase on fearfulness. They could be less curious with age, the test was repeated and they now the NO was no longer novel, their life experience changed the perceived novelty of a NO, etc.

L291-292 Please start the sentence with “This indicates that …”

L293-301 there is a need for some references here

L303-305 The wording suggests a causality that is not proven. ‘As soon as’ indicates that the feather damage was first and the fearfulness followed. If anything, the literature suggests the opposite which you show in the following paragraph.

L324 change ‘confirm’ to ‘indicate’

L331-333 confounded by strain?

L334-335 this result is contrary to your graph!

L347 I would suggest extending on the future direction a little bit. There are some more ideas in the discussion that could be added here.

Conclusion

L352-353 again, do not imply causality. We do not know if and increase in feather pecking and cannibalism made the birds more fearful or the other way around. You found an association.

L354-355 But you found a farm type effect!

Minor comments:

L44 change status to state

L45-48 this seems unnecessarily detailed. I would suggest reducing and merge with the previous or following sentence.

L52 change ‘of’ animals to ‘in’ animals

L54 ‘accompanied’ does not appear to be the right word. Were you going for ‘studied’, or ‘investigated’ or maybe ‘followed’?

L98-99 Reword this sentence: ‘…in accordance with standard industry procedures, e.g., vaccinations against …’

L100 I suggest changing ‘imply’ to ‘include’ or ‘impose’ instead.

L105 conventional housing in laying hens is often understood as caged housing. I would suggest deleting the word.

L106 ‘wintergardens’ are not common everywhere (mostly exclusive to Europe), please define it for the readers on other continents.

L109-110 Rephrase this sentence as it is unclear whether the hens stayed in this housing for 62-95 weeks or until they were 62-95 weeks old.

Figure 1 do you have a photo where the NO is in action? That could be a nice addition.

L118-120 instead of listing the other variables collected at each visit, add the experimental procedure to the figure caption: NO was placed…, birds were counted…, NO was removed when…

122 What did the observer do during the 5 minutes before the test?

L124 onwards: what did the observer do during the test, where were they observing from?

L131 What does this mean? Be more specific please.

L132 This paragraph does not require its own header and should be moved to the beginning of the M&M at line 99 after ‘Marek Disease.’ and before ‘In compliance…’.

L143 Please explain the variation.

L146 HSc should have a capital S

L149 For the future, you can turn off the lights and use night vision goggles or blue light headlamps to catch hens without stressing them out.

L181 the punctuation seems off. Please check with Grammarly or a similar software.

L183 Not sure if the abbreviation SD was introduced?

Figure 2 and all following figures: Please use the significant indications that are common: *= p<0.05, **= p<0.01, ***= p<0.001.

Figures 5 and 6 can be combined into a single figure with 2 graphs.

L168-270 I would suggest that you re-phrase this sentence to say that in comparison to previous studies you involved a large number of commercial flocks.

L277 change ‘due to significantly’ to ‘as indicated by a’

L284 change ‘early’ to ‘earlier’

L289 ‘open field’? you did not do an open field test. Be carful with wording that has a loaded meaning to the audience.

L309 unclear what the ‘Therefore’ is in reference to.

L324 add ‘of a NO’ to ‘… fearful than brown hens‘.

L326 add the reference right after ‘Jones’.

L336 change ‘make’ to ‘draw’.

L348 there is an awkward repetition in this sentence. Reword to is has the definition of latency right after the word

There are a few awkward sentences, some problems with punctuation, and a few words that appear incorrect. I have indicated which words should be replaced and offered alternatives. I would suggest using Grammarly (personal recommendation as a non-native speaker) or similar software.

Author Response

Dear Reviewer,

Yours sincerely,

Jennifer Hüttner

Reviewer 2 Report

Review animals

The ethical approval is missing.

This is a small scale field study linking the behavior during a novel object test in the barn with feather pecking and cannibalism with interesting results. However, there are several problems with this study. My main concern is that it is not clear if the statistical analyses are valid. The model of the statistical analyses and the type of analyses are not given. It is unclear how the subsequent visits were analyzed, as a repeated factor? For some categories, sample sizes were very small like 2 or 3 flocks. Likely, factors were confounded, e.g. I doubt that the 3 organic flocks consisted of white and brown hybrids and even then, you would be left with just one organic flock of a particular color. The same is true for the different flock sizes with categories consisting of 2 or 3 flocks. The authors have to be much more cautious with the interpretation of the results.

 Another problem is the interpretation of the behavior tests. When you always use the same object, the object is not novel anymore. Please discuss the implications of this. Normally, other objects are used at subsequent visits. The term 'novel object' should only be used at the first visit and then the term 'object' should be used. Additionally, if the observer stood near the object, fear of humans and fear of an object cannot be separated and comparisons with existing literature about NOT might not be valid.

As the authors rightly explain in the introduction, the NOT measures the conflicting behaviors exploration and avoidance due to fear. For that reason, the outcome of the NOT is influenced by both factors and not just fear. The authors should consider this in the discussion and conclusion of this study and discuss the possibility that the hens differed in their desire to explore the object and not fear.

Line 21: indicate

Line 33: What are barn farms? Don't all farms have barns?

Line 33ff: You report a statistically significant difference and then you declare that there was no effect on fearfulness. How does this fit?

Additionally, this seems to be an observational study (you did not experimentally control access to outdoor space). So you should drop the word 'affect' and use 'to be associated' instead.

Line 54: What does accompanied mean?

Lines 61f: The subject of the sentence is missing.

Line 80: were identified.

Lines 121ff: What was the distance between the NOT and the observer? Do you think the presence of the human influenced the outcome of the test? Did you test the fear of humans or the fear of the novel object?

Line 143: Normally (MUD, Welfare Quality) at least 50 hens are investigated. I doubt that 25 is a sufficient number of animals. How did you catch the animals? Did you turn-off / dim the lights? Otherwise, it is not possible to get a representative sample because all fearful hens will be gone.

Line 153: What was the intra/inter-observer reliability between Vsc and HSc?

Methods: Were the observers blinded during the NOT? If not, how can you assure that the appearance of the plumage did not influence the outcome of the NOT? Even when the plumage was assessed later, the presence of plumage damage is obvious as soon as you enter a barn.

Line 167: Please give the statistical model how you analyzed the data. What were your fixed and random factors? Was visit a repeated factor? Did you calculate contrasts? With which method?

Lines 183ff: Did you first compare the barn farms with all other farms? If not, you should write free-range but not organic farms because organic farms must all have a free range.

Line 185: I think a sample size of 3 for organic farms is too low to analyze this as a factor.

Line 187f: This belongs to the methods.

Figure 2: I would not draw a bar plot of 3 values. This is not meaningful.

Figure 3: I would not talk about a novel object on visits 2 to 4. You can call it an object because it is not novel.

I don't understand the text for Figure 4. As I see it, the smaller the flock the more hens were around the NOT. In the text it is described the other way around.

Line 223: Since this is an observational study you should not use the term 'influence'. You can only talk about associations.

Figure 5: Please give the sample size of the categories.

Line 242: Again, drop 'influence'.

Line 244: were instead of was. In those flocks there were more hens around the novel object ….

Line 247: were instead of was.

Figure 6: Please give the sample sizes of the different categories.

Line 269: I don't understand the second part of the sentence.

Lines 274ff: There is an alternative explanation: Maybe the hens were less explorative with increasing age instead more fearful.

Line 306: Maybe birds are more fearful because they have the risk of being pecked. The causation between feather-pecking and fear is unknown.

Line 323: I am not sure, maybe brown hens are more curious than white hens.

Paragraph 334ff: This result was contradicting the cited studies. Maybe the environment was more boring for hens without outdoor access so they went to the novel object instead. This would mean that the NOT did not only measured fear but also exploration.

Line 351ff: I disagree with your first sentence of the conclusion. It is too strong.

The English must be corrected as included in the comments.

Author Response

Dear reviewer,

Yours sincerely,

Jennifer Hüttner

Reviewer 3 Report

The present study aimed to assess the fearfulness of laying flocks using the novel object test (NOT) under practical conditions and investigate the impact of housing and welfare factors. The study observed 16 flocks in Germany, comprising barn, free-range, and organic systems. The plumage and integument condition of 50 birds from each flock were evaluated at multiple time points during the laying period, alongside the performance of the NOT. The results revealed a correlation between fearfulness and increased feather damage and cannibalism. Age and flock size were identified as influencing fearfulness, with older hens and larger flocks displaying more fear-related behaviors. Hens of white genetics exhibited higher fearfulness levels compared to brown hens. Surprisingly, barn farms had a higher number of hens gathered around the novel object than free-range farms, suggesting that outdoor access may not consistently impact flock fearfulness.

Overall, the findings of this study shed light on the relationship between fearfulness and various factors in laying flocks. The results contribute to our understanding of welfare implications and highlight the influence of genetic and environmental factors on fear-related behaviors.

I would like to suggest starting the introduction of your paper with a general negative comment on farm welfare in intensive animal production systems across species. For example beef cattle (see and cite https://doi.org/10.1016/j.rvsc.2023.03.008), beef horses (see and cite: 10.3390/ani12141740), dairy goat (see and cite: 10.3390/ani13050797), rabbit (see and cite: 10.1080/1828051X.2020.1827990). This can help set the context and emphasize the importance of studying fearfulness and welfare-related parameters in laying flocks. By briefly addressing the broader concerns surrounding farm welfare, you can engage readers and highlight the significance of your research in addressing these issues.

Introducing the topic with a broader perspective on farm welfare challenges can enhance the relevance and significance of your study. It will also provide a stronger motivation for your specific research objectives and contribute to the overall understanding of welfare issues in intensive animal production.

I would like to suggest including additional information in the Materials and Methods section of your paper. Specifically, please provide more details regarding the genetic type of the laying flocks, including the specific breeds or genetic lines used. Additionally, please elaborate on the different types of flocks mentioned in line 91, such as barn, free-range, and organic flocks, and provide a brief description of their housing and management systems.

Furthermore, it would be beneficial to include information about the breeding type employed in your study, particularly whether the flocks were kept in cages or if any other specific breeding methods were used.

Additionally, please provide a detailed description of the feeding regimen and management practices implemented for the flocks during the laying period. This information will enhance the clarity and reproducibility of your study and allow readers to better understand the context in which the fearfulness and welfare parameters were assessed.

What do you mean with term barn? Aviary system? please be more specific

I appreciate the valuable insights provided in your study on the fearfulness of laying flocks and its association with housing and welfare parameters. However, I would like to suggest considering the inclusion of additional parameters in your investigation.

Firstly, monitoring mortality rates can provide important information about the overall health and well-being of the laying flocks. By assessing mortality rates at different stages of the laying period, you can gain insights into the potential impacts of fearfulness on flock survival and identify any patterns or correlations with other factors.

Secondly, evaluating the occurrence and severity of lesions in the birds' plumage and integument can provide a more comprehensive understanding of their welfare status. Lesions can indicate potential issues related to housing conditions, management practices, or flock behavior. Assessing lesion scores or categorizing the presence and severity of specific types of lesions would provide valuable data for further analysis and discussion.

Lastly, incorporating measurements of body condition, such as body weight or body condition scores, can contribute to a more holistic assessment of the laying flocks' welfare. Body condition can reflect nutritional status, stress levels, and overall health, and it can provide additional insights into the potential impacts of fearfulness on the birds' physiological well-being.

I have thoroughly reviewed your study on the fearfulness of laying flocks and its relationship with housing and welfare parameters. While your findings provide valuable insights, I would like to suggest incorporating a discussion of relevant studies that have investigated laying hens under different freedom conditions and plumage conditions. By comparing your results with existing literature, you can enhance the context and implications of your findings.

Specifically, I recommend considering studies that have examined the fearfulness of laying hens in various housing systems, such as cage-free, free-range, or enriched colony systems. Exploring how fearfulness levels in your study compare to those reported in different housing systems would provide valuable insights into the influence of housing conditions on laying hen welfare and fear-related behaviors.

Additionally, considering studies that have investigated the impact of plumage conditions on fearfulness would be beneficial. Assessing the fear responses of hens with intact plumage compared to those with feather damage or plumage abnormalities can shed light on the relationship between fearfulness, plumage condition, and potential welfare challenges.

Please see and cite: 10.3390/ANI12182307/S1

line 322-323: this is strange because with genetic types usually have more nervous behavior

Please ensure that each reference follows the correct citation format, including the accurate arrangement of author names, correct use of punctuation, consistent capitalization, and inclusion of all necessary information such as article titles, journal names, volume and issue numbers, page ranges, and DOIs (if applicable).

A well-formatted and accurate reference list will greatly enhance the overall quality and professionalism of your paper. Thank you for your attention to this matter.

Author Response

(The authors gave the same response as above.)

Round 2

Reviewer 2 Report

Review animals

The manuscript has been improved but it still requires a lot of changes.

Sometimes you use the term genetics and sometimes feather color. Feather color is more specific so I would use this term throughout the manuscript.

In general, the writing is poor and an English editing service should correct the writing.

Title: Associations … with instead of on.

Line 20f: The sentence has a weird grammar.

Line 30f: Drop ‘as well’.

Line 84: … association … with …

Line 125f: How can you study age effects when you averaged the four visits?

Line 137f: … maximum of

Line 135: What does the hyphen mean?

Line 163: On average …

Line 207: Not clear, were all these variables random factors? Or does this refer only to the genetics? If yes, replace genetics by plumage color. Then it is more clear.

Line 234: Associationbetween is two words.

Line 296: Replace indicate by investigate or explore.

Line 298: a instead of an.

Line 298: The word ‘under’ does not fit, rewrite.

Line 304: Drop ‘with’ after ‘involved’.

Line 329ff: This sentence is awkward and should be rewritten.

Line 332: … taking into account …

Line 339: an association …

Line 347: Can be? Please replace with is correlated or was correlated.

Line 369: Insert space between NO and thus.

Line 400: Were instead of are associated … . Every result of this study should be written in the past tense and previously published results in the present tense.

The English has to be improved substantially. Please use an English editing service.

Author Response

Dear reviewer,

Yours sincerely,

Jennifer Hüttner (on behalf of all co-authors)

Reviewer 3 Report

after the revisions the paper improved a lot, I endorse the publication, well done authors

Author Response

Dear Reviewer,

thank you for your Report. We would like to thank you for the comments and suggestions you have made previously.

Yours sincerely,

Jennifer Hüttner (on behalf of all co-authors)